The Cryosphere

#### Response of freeze-thaw processes to experimental warming in the 1 permafrost regions of the central Qinghai-Tibet Plateau 2 3 Shengyun Chen<sup>1, 2, +, \*</sup>, Wenjie Liu<sup>3, 1, +</sup>, Qian Zhao<sup>1, 7</sup>, Lin Zhao<sup>4</sup>, Qingbai Wu<sup>5</sup>, Xingjie Lu<sup>6</sup>, 4 Shichang Kang<sup>1</sup>, Xiang Qin<sup>1</sup>, Shilong Chen<sup>2</sup>, Jiawen Ren<sup>1</sup>, and Dahe Qin<sup>1</sup> 5 6 7 <sup>1</sup>Qilian Shan Station of Glaciology and Ecologic Environment, State Key Laboratory of Cryosphere Science, Cold 8 and Arid Regions Environmental and Engineering Research Institute, Chinese Academy of Sciences, Lanzhou, 9 730000 China 10 <sup>2</sup>Key Laboratory of Adaptation and Evolution of Plateau Biota, Northwest Institute of Plateau Biology, Chinese 11 Academy of Sciences, Xining, 810001, China 12 <sup>3</sup>College of Environment and Plant Protection, Hainan University, Haikou Hainan 570228, China 13 <sup>4</sup>Cryosphere Research Station on the Qinghai-Tibetan Plateau, State Key Laboratory of Cryosphere Science, Cold 14 and Arid Regions Environmental and Engineering Research Institute, Chinese Academy of Sciences, Lanzhou, 15 730000, China 16 <sup>5</sup>Beiluhe Observation and Research Station on Frozen Soil Engineering and Environment in Qinghai-Tibet Plateau, 17 State Key Laboratory of Frozen Soil Engineering, Cold and Arid Regions Environmental and Engineering 18 Research Institute, Chinese Academy of Sciences, Lanzhou, 730000, China 19 <sup>6</sup>CSIRO Oceans and Atmosphere, Aspendale, Victoria, 3195, Australia 20 <sup>7</sup>University of Chinese Academy of Sciences, Beijing, 100049, China 21 22 + These authors contributed equally to this work 23 \* Correspondence to: Shengyun Chen (sychen@lzb.ac.cn) 24 25 Abstract. Assessing quantitatively effect of climate warming on freeze/thaw index (FI/TI), soil 26 freeze-thaw processes and active layer thickness (ALT) is still lacking in the permafrost regions of 27 the Qinghai-Tibet Plateau (QTP) until now. Experimental warming was manipulated using open 28 top chambers (OTCs) in alpine swamp meadow and alpine steppe ecosystems in the permafrost 29 regions of the central QTP during 2009-2011. Under OTCs treatment, air temperature (Ta) 30 significantly increased in the daytime and decreased in the nighttime, diurnal and annual Ta range 31 significantly enhanced, and mean annual Ta increased by 1.4 °C. Owing to the experimental 32 warming, mean annual soil temperature at the depths from 5cm to 40cm was increased by 0.2 $\sim$ 33 0.7 °C in alpine swamp meadow and 0.3 ~ 1.5 °C in alpine steppe. Mean annual soil moisture 34 content at 10 cm depth decreased by 1.1 % and 0.8 %, and mean annual soil salinity at 10 cm 35 depth significantly increased by 0.3 g $L^{-1}$ and 0.1 g $L^{-1}$ in alpine swamp meadow and alpine steppe, respectively. Further, FI was significantly decreased by 410.7 °C d while TI was significantly 36 37 increased by 460.7 °C d. Likewise, the onset dates of shallow soil thawing at 5-40 cm depths were 38 advanced by 9 days and 8 days while the onset dates of freezing were delayed by 10 days and 4 39 days in alpine swamp meadow and alpine steppe, respectively. Moreover, soil frozen days were

40 significantly decreased by 28 days and 16 days, but thawed days were increased by 18 days and 6

days, and frozen-thawed days were significantly increased by 10 days and 10 days in alpine
swamp meadow and alpine steppe, respectively. Furthermore, ALT would be significantly
increased by ~6.9 cm and ~19.6 cm in alpine swamp meadow and alpine steppe ecosystems,
respectively.

5

# 6 1 Introduction

7 The fifth assessment report (AR5) of the Intergovernmental Panel on Climate Change (IPCC) 8 indicated that the global average surface temperature had increased by  $0.85 (0.65 \sim 1.06)$  °C over 9 the period of 1880 to 2012, and the change for the end of the 21st century would be likely to exceed 1.5 °C than the average from 1850 to 1900 (IPCC, 2013). Likewise, the mean annual 10 temperature in the Qinghai-Tibet Plateau (QTP) has increased by 0.2 °C per decade over the past 11 50 years, and is expected to increase 2.6-5.2 °C by 2100 under the A2 and B1 scenarios (Chen et 12 13 al., 2013). The warming results ineluctably in a notable change in cryosphere. The QTP is the 14 highest plateau and considered as the "Third Pole" on Earth, with an average elevation exceeding 4000 m a.s.l. and an area of about  $2.5 \times 10^6$  km<sup>2</sup> (Yang et al., 2014). Permafrost, as one of the 15 major cryospheric components, underlies approximately  $1.05 \times 10^6$  km<sup>2</sup> (glaciers and lakes 16 17 excluded) in the QTP (Ran et al., 2012). Meanwhile, as a product of cold climate, it is extremely 18 sensitive to climate change (Chen et al., 2012a). Previous studies in the QTP showed that climate 19 warming has caused significant changes in freezing/thawing index (FI/TI) and soil freeze-thaw 20 processes (SFTPs) of the active layer in the permafrost regions (Jiang et al., 2008; Zhao et al., 21 2008; Li et al., 2012a; Li et al., 2012b; Wu et al., 2013; Jin et al., 2015; Wang et al., 2015). 22 Furthermore, permafrost has also experienced widespread degradation including the rise of ground 23 temperature and the increase of active layer thickness (ALT) over the past several decades (Wu 24 and Zhang, 2008; Wu and Zhang, 2010; Li et al., 2012a; Wu et al., 2012; Wu et al., 2015). These 25 would unavoidably lead to deterioration of alpine grassland ecosystem, positive feedback of carbon cycling and damage of engineering infrastructure and so on (e.g., Schuur et al., 2009; 26 27 Koven et al., 2011; Chen et al., 2012a; Hicks Pries et al., 2016).

28 The annual FI/TI, as a general indicator of climate change in the cold regions, is useful for 29 predicting permafrost distribution and is an important parameter for engineering design 30 (Frauenfeld et al., 2007). In addition, variation in the near-surface SFTPs can be used as an effective and reliable indicator of permafrost change (Li et al., 2012b). SFTPs are the repeated 31 32 freeze-thaw physical processes, which are resulted from diurnally or seasonally thermal changes 33 following with water phase transitions in the topsoil up to a certain depth of soil. The processes 34 are common in some temperate, most high-altitude and middle-high-latitude regions (Grogan et 35 al., 2004). Zhang et al. (2003) estimated that approximately 57% of land surface experienced SFTPs annually in the Northern Hemisphere. SFTPs can be significantly affected by local 36 37 environmental factors including climate, microtopography, vegetation and hydrology (Minke et 38 al., 2009; Guglielmin et al., 2012; Wang et al., 2012). On the contrary, SFTPs also have a strong,

profound and extensive influence on vegetation productivity, soil structure, soil hydrothermal

dynamics, energy-water-carbon exchanges between land surface and atmosphere, as well as

engineering infrastructure stability and so on (e.g., van Bochove et al., 2000; Grogan et al., 2004;

Qi et al., 2006; Zimov et al., 2006; Kreyling et al., 2008; Wang et al., 2009; Zhang and Sun 2011;

Tagesson et al., 2012; Wang et al., 2012; Wang et al., 2014; Hu et al., 2015).

A few studies have reported the effect of short-term warming on STFPs of active layer using 7 open-top chambers (OTCs) facility in alpine swamp meadow of the QTP (Liu et al., 2012a; Wang 8 and Wu, 2013). However, a quantitatively systematic research is still lacking about response of 9 FI/TI, soil hydrothermal feature and SFTPs, as well as ALT to climate warming in alpine grassland ecosystem in the permafrost regions of the QTP so far. Therefore, the main objectives of this study 10 11 were to detect variations of Ta and shallow soil hydrothermal characteristics, and to especially 12 focus on revealing the response of freeze-thaw processes including FI/TI and SFTPs in two typical 13 alpine grassland ecosystems to OTCs experimental warming in the permafrost regions of the 14 Beiluhe Basin of the central QTP during 2009-2011. Finally, ALT changes were further estimated 15 under the warming scenarios. The work will be helpful to evaluate the stability of Qinghai-Tibet Railway/Highway and estimate the release of old carbon, owing to changing freeze-thaw 16 17 processes and thawing permafrost under the future climate warming in the QTP.

### 18 2 Materials and methods

### 19 2.1 Study area and site description

20 The Beiluhe is one of the primary tributaries of Tongtian River in the upper reaches of Yangtze 21 River. The Basin belongs to typical plateau continental semi-arid climate with rarefied air, low air 22 temperature, little but concentrated rainfall, low pressure and strong ultraviolet radiation (Chen et 23 al., 2012b; Zhang et al., 2014). And it is mainly covered by alpine grassland, which occupies 94% 24 of the total area of about 7949 km<sup>2</sup>, and the main alpine grassland types are alpine steppe (~ 4300 25 km<sup>2</sup>), alpine meadow (~ 3090 km<sup>2</sup>) and alpine swamp meadow (~ 96 km<sup>2</sup>) (Liu et al., 2014). The 26 study area is located in the upstream regions of Beiluhe Basin of the central QTP (Fig. 1). Based 27 on the data of meteorological station from 2009 to 2011, mean annual air temperature (Ta), 28 relative humidity (RH), precipitation (P), net radiation and wind velocity were -3.2 °C, 53.3 %, 29 401.4 mm, 83.7 W m<sup>-2</sup> and 4.5 m s<sup>-1</sup>, respectively. The warmest and coldest months were July and 30 January, the maximum and minimum precipitations appeared in July and December, respectively 31 (Fig. 2). The region was mainly underlain by continuous and ice-rich permafrost (Wen et al., 2014; 32 Luo et al., 2015). 33 Two sites of typical alpine grasslands including degraded alpine swamp meadow (BLH1 site)

and alpine steppe (BLH3 site) were selected nearby the Beiluhe Observation and Research Station
 on Frozen Soil Engineering and Environment in Qinghai-Tibet Plateau, Cold and Arid Regions
 Environmental and Engineering Research Institute, Chinese Academy of Sciences (Beiluhe
 Station; 34°51.26'N, 92°56.35'E, *Alt.*: 4659m; Fig. 1). In addition, two sites are adjacent to

1 Qinghai-Tibet Railway and Highway, which BLH1 site is about 35m apart from railway and 2 BLH3 site is approximately 130m apart from highway. Vegetation coverage of BLH1 and BLH3 3 sites was about 87% and 38%, respectively. Dominant plants were Kobresia tibetica, K. robusta 4 and Carex moorcroftii in BLH1 site, Stipa purpurea, Saussurea arenaria and C. moorcroftii in 5 BLH3 site (Chen, 2010; Zhang et al., 2014). According to the classification of IUSS Working 6 Group WRB, the soil types of BLH1 and BLH3 sites were Calcic Kastanozem and Gelic 7 Arenosols, respectively (Liu et al., 2014). Sandy and clay fractions of surface soil were 60.6% and 8 9.7% in BLH1 site, 68.8% and 7.6% in BLH3 site, respectively. ALT of BLH1 and BLH3 sites 9 were about 200cm and 220cm, respectively, on the basis of pitting measurement in late September 2009. 10

#### 11 2.2 Experimental design and data collection

12 Each site of 30 m  $\times$  30 m was protected by wire fence to prevent disturbance from humans or 13 animals. Within each site, five circular plots (1.5 m in diameter) were laid out and spaced 10 m 14 apart, then five replicated conical OTCs as the experimental warming treatment were placed 15 within plots in late October 2008 and remained throughout the whole period, respectively. Each OTC was constructed by a fiberglass material of Sun-Lite HP® (Solar Components Corporation, 16 17 Manchester, NH, USA) 1.0mm thickness, which was the basal diameter of 1.48m and the height of 0.40m shaped like truncated cones with inwardly inclined sides (60° with respect to the 18 19 horizontal). The material and design of OTCs were fully in accordance with standards of the 20 International Tundra Experiment (ITEX) (Molau and Mølgaard, 1996). In order to avoid edge 21 effects of chambers, parallel control plots (CPs) as the ambient environment were established 22 2.5m apart from their adjacent OTCs warming plots. This OTCs facility has been commonly 23 employed to study the effect of climate warming on alpine grassland ecosystem in the QTP (e.g., 24 Klein et al., 2004, 2005).

Within OTC and CP at each site, Ta was measured with HMP45C probes protected by Gill radiation shield (Vaisala, accuracy of  $\pm 0.5$  °C). The probes were centered in 15 cm height above 26 27 the ground surface. Soil temperature (Ts) at 5 cm, 10 cm, 20 cm, 30 cm and 40 cm depths below 28 the soil surface was measured using thermistor sensors (accuracy of  $\pm 0.05$  °C) made by the State 29 Key Laboratory of Frozen Soil Engineering, Cold and Arid Regions Environmental and 30 Engineering Research Institute, Chinese Academy of Sciences. Soil moisture content (SM) and 31 soil salinity (SS) at 10cm depth were measured by Hydra Probe sensors (Stevens) with the accuracies of  $\pm 3.0\%$  and  $\pm 0.03$  g L<sup>-1</sup>, respectively. All probes and sensors were connected with 32 CR1000-XT datalogger (Campbell), and data were automatically recorded every hour (Beijing 33 34 Standard Time).

#### 35 2.3 Data analysis

36 The number of soil frozen days, thawed days and frozen-thawed days were calculated by those

days with the diurnal maximum Ts being  $\leq 0.0$  °C, the diurnal minimum Ts being > 0.0 °C and the

1diurnal maximum Ts being > 0.0 °C but the diurnal minimum Ts being  $\leq$  0.0 °C, respectively. The2onset date of soil thawing or freezing was determined by the first day of three consecutive thawing3or freezing days based on mean daily Ts. Mean seasonal values for spring, summer, autumn and4winter were calculated through mean monthly values of Mar.-May, Jun.-Aug., Sep. -Nov. and5Dec.-Feb., respectively. Paired T-test was used to examine the statistical significance of variables6between OTCs and CPs in SPSS16.0. The FI/TI were calculated using the following equations (1)7and (2) by referring to Franuenfeld et al. (2007) and Jiang et al. (2008):

$$FI = \sum_{i=1}^{N_F} |Ti|, \ Ti \leqslant 0.0 \,^{\circ} \mathbb{C}$$

$$\tag{1}$$

$$TI = \sum_{i=1}^{N_T} |Ti|, \ Ti > 0.0 \ ^{\circ} \mathrm{C}$$

$$\tag{2}$$

where *T*i represented the mean daily temperature between daily maximum and minimum *T*a in 15cm height above the ground.  $N_F$  and  $N_T$  corresponded to those days when temperature is below (or equal to) and above 0.0 °C, respectively. In addition, ALT was also calculated using the empirical equation (3) explained by Li et al. (2012a):

$$ALT = a \cdot Ta + b \cdot CTN + c \cdot CTP + d \tag{3}$$

where a, b, c and d were regression coefficients equaled to -7.00, 0.07, 0.04 and 272.88, respectively. *Ta*, CTN and CTP represented summer *Ta*, annual negative cumulative and positive

17 cumulative *T*s at 5cm depth, respectively.

### 18 3 Results

8

9

14

## 19 **3.1 Variation in air temperature (Ta)**

20 From 2009 to 2011, diurnal Ta varied from -8.9 °C to 3.8 °C in CPs and from -9.4 °C to 8.4 °C in 21 OTCs. Diurnal Ta in OTCs was significantly higher from 9:00 to 20:00 (P

1 Diurnal fluctuation of Ts gradually reduced with the increase of profile depth under OTCs and CPs 2 treatments of BLH1 and BLH3 sites, which was greater at BLH3 site than at BLH1 site, especially 3 at 5-20 cm depths. Owing to OTCs experimental warming, diurnal Ts at the depths of 5 cm, 10 cm, 20 cm, 30 cm and 40 cm maximally increased by 2.9 °C, 1.5 °C, 0.4 °C, 0.4 °C and 0.4 °C at BLH1 4 site and by 3.3 °C, 2.3 °C, 0.9 °C, 0.3 °C and 0.5 °C at BLH3 site, respectively. The warming 5 amplitudes at 5-20 cm depths were higher at BLH3 site than at BLH1 site, and the warming over 6 7 twenty-four hours occurred at 5 cm depth of BLH3 site (Fig. 4). In addition, mean daily Ts 8 maximally increased by 2.3 °C and 4.1 °C, 2.1 °C and 3.1 °C, 1.3 °C and 1.9 °C, 1.8 °C and 1.1 °C, 9 1.5 °C and 1.7 °C at 5 cm, 10 cm, 20 cm, 30 cm and 40 cm depths of BLH1 and BLH3 sites, respectively (Fig. 7). Further, mean monthly Ts basically increased except Jul.-Sep. at 5 cm depth 10 11 and Jun.-Oct. at 10-40 cm depths at BLH1 site, and over twelve months at 5 cm depth and except 12 Jun.-Sep. at 10-40 cm depths at BLH3 site. Meanwhile, the maximum increases of mean monthly 13 Ts at 5 cm, 10 cm, 20 cm, 30 cm and 40 cm depths were 1.3 °C (May), 1.2 °C (Feb.), 1.2 °C (Jan.), 14 1.5 °C (Jan.) and 1.4 °C (Jan.) at BLH1 site, as well as 2.8 °C (Apr.), 2.3 °C (Dec.), 1.4 °C (Dec.), 15 0.6 °C (May) and 0.9 °C (Dec.) at BLH3 site, respectively. (Fig. 4). Mean seasonal Ts significantly increased in spring, autumn and winter at 5 cm depth, and in spring at 30-40 cm depths and in 16 17 winter 20-40 cm depths of BLH1 site, as well as in autumn and winter at 5-10 cm depths, in 18 autumn at 40 cm depths, in winter at 20 cm and 40 cm depths of BLH3 site (P < 0.01 or 0.05). 19 Furthermore, maximum increases of mean seasonal Ts at the depths from 5 cm to 40 cm were 1.1 20 °C and 2.4 °C (spring), 0.9 °C and 2.1 °C (winter), 1.0 °C and 1.0 °C (winter), 1.0 °C (winter) and 21 0.5 °C (spring), 0.8 °C and 0.8 °C (winter) at BLH1 and BLH3 sites, respectively. Of course, 22 increases of mean seasonal Ts at 5-10 cm depths were significantly lower at BLH1 site than those 23 at BLH3 site in autumn and winter (P < 0.05) (Table 1). Overall, mean annual Ts at the depths from 24 5 cm to 40 cm increased by  $0.2 \sim 0.7$  °C at BLH1 site, and by  $0.3 \sim 1.5$  °C at BLH3 site (P<0.05), 25 as well as increases of Ts at 10cm were significantly higher in BLH3 site than those at BLH1 site 26 (P<0.05) (Table 1). 27 On the basis of Fig. 5, mean daily SM and SS at 10 cm depth changed by -7.8 % (May 4th) ~ 28 12.1 % (Nov. 4<sup>th</sup>) and 0.1 g L<sup>-1</sup> (Feb. 4<sup>th</sup>) ~ 0.7 g L<sup>-1</sup> (Jul. 6<sup>th</sup>) at BLH1 site, and by -1.0 % (Oct.

23rd) ~ -0.5 % (Dec. 6th) and -0.1 g L<sup>-1</sup> (Jan. 27th) ~ 0.1 g L<sup>-1</sup> (Aug. 13rd) at BLH3 site because of 29 30 OTCs warming, respectively. Furthermore, mean monthly SM decreased by 0.3 ~ 4.6 % in other 31 months except Jul. and Nov. at BLH1 site and by  $0.5 \sim 1.0$  % at BLH3 site, as well as mean 32 monthly SS increased by  $0.1 \sim 0.5$  g L<sup>-1</sup> over twelve months at BLH1 site and by  $0.04 \sim 0.1$  g L<sup>-1</sup> 33 in other months except Jan., Feb. and Dec. at BLH3 site (Fig. 6). Moreover, mean seasonal SM 34 significantly decreased in winter at BLH1 site, in each season at BLH3 site (P

- summer, autumn and winter (P<0.05; Table 1). However, mean annual SM decreased by 1.1 % at
- BLH1 site and by 0.8 % at BLH3 site (P<0.01; Table 1). Mean annual SS significantly increased
- by 0.3 g L<sup>-1</sup> and 0.1 g L<sup>-1</sup> at BLH1 and BLH3 sites, respectively, which was significantly higher at
- BLH1 site than that at BLH3 site (*P*<0.05; Table 1).

**3.3** Variations in freezing/thawing index (FI/TI), soil freeze-thaw processes (SFTPs) and 6 active layer thickness (ALT)

Under CPs treatment during 2009 ~ 2011, freezing index (FI) and thawing index (TI) varied from 8 1814.4 °C d to 1980.5 °C d and from 820.4 °C d to 955.8 °C d, with average values of 1873.4 °C d 9 and 886.8 °C d, respectively. However, the average values of FI and TI were 1462.7 °C d and 1347.5 °C d, with variations of 1388.4 ~ 1580.0 °C d and 1271.0 ~ 1422.1 °C d under OTCs 11 treatment, respectively. Of course, FI significantly decreased by 410.7 °C d, but TI obviously 12 increased by 460.7 °C d owing to experimental warming (*P*<0.01; Table 2).</p>

According to profile variations of Ts from 2009 to 2011 in Fig. 7, Ts gradually increased 13 14 during freezing stage while decreased during thawing stage, with the increase of profile depths 15 from 5 cm to 40 cm. Meanwhile, the onset dates of soil thawing/freezing at the depths from 5 cm to 40 cm were on Apr. 19th ~ May 20th / Oct. 23 rd ~ Nov. 9th and Apr. 8th ~ May 7th / Oct. 31 st ~ 16 Nov. 16<sup>th</sup> at BLH1 site, as well as Apr. 11<sup>st</sup> ~ Apr. 30<sup>th</sup> / Oct. 24<sup>th</sup> ~ Nov. 7<sup>th</sup> and Mar. 26<sup>th</sup> ~ Apr. 17 26<sup>th</sup>/Oct. 31<sup>st</sup> ~ Nov. 9<sup>th</sup> at BLH3 site under CPs and OTCs treatments, respectively. Obviously, 18 19 the onset dates of thawing were advanced by 5 ~ 13 days and 3 ~ 16 days with averages of 9 days 20 and 8 days, but the onset dates of freezing were delayed by  $7 \sim 15$  days and  $1 \sim 7$  days with mean 21 values of 10 days and 4 days at BLH1 and BLH3 sites owing to OTCs warming, respectively. 22 Furthermore, soil frozen days gradually increased, frozen-thawed days decreased, and thawed 23 days first increased and then decreased from 5 cm to 40 cm under two treatments. Owing to OTCs 24 warming, frozen days of shallow soil at 5-40 cm depths decreased by  $21 \sim 51$  days and  $6 \sim 39$ 25 days with averages of 28 days and 16 days, and thawed days increased by 14 ~ 21 days and 5 ~ 6 days with averages of 18 days and 6 days, as well as frozen-thawed days increased by 1 ~ 37 days 26 27 and 1 ~ 33 days with averages of 10 days and 10 days at BLH1 and BLH3 sites, respectively. 28 Moreover, frozen days generally decreased while thawed and frozen-thawed days increased, 29 especially variations of frozen and frozen-thawed days were significant at the depths of 5 cm and 30 5-40 cm (P

1 preceding decade since 1850, and the warming of global surface temperature would continue 2 beyond 2100 (IPCC, 2013). Likewise, QTP has experienced a universal and significant warming 3 (Wu and Zhang, 2008; Li et al., 2010; Wu et al., 2013), the recent warming rate has been greater 4 than those for the northern hemisphere, the southern hemisphere and the world as a whole 5 (Trenberth et al., 2007), and the future Ta is likely continued to increase (Chen et al., 2013). In 6 order to study response processes of terrestrial ecosystem to climate warming, in-situ experimental 7 warming in the field has become one of the main methods with a variety of facilities including 8 OTCs, greenhouses, soil heating cables and infrared radiator and so on (Wan et al., 2002). The 9 effect of passive OTCs facility on microclimate mainly depends on variables including solar 10 radiation and wind speed (Klein et al., 2005). Of course, these variables can be inevitably affected 11 by OTCs, and the variable most influenced is wind speed (De Boeck et al., 2012). Further, the 12 wind speed reduction under OTCs treatment will lead to large Ta differences owing to the passively-ventilated shield (Tarara, 2007). 13

14 To data, OTCs facility has been generally used in the remote and no-electric regions 15 including the Arctic, Antarctica and QTP. In this study, the short-term OTCs experimental warming during 2009-2011 was conducted in two typical alpine grassland ecosystems including 16 17 alpine meadow swamp meadow (BLH1 site) and alpine steppe (BLH3 site) in the permafrost 18 regions of the central QTP. Previous studies of OTCs experimental warming have reported that Ta 19 variations showed daytime warming while nighttime cooling, and increases of diurnal temperature 20 range and mean daily Ta, as well as higher enhancement of mean seasonal Ta in summer than in 21 winter (Kennedy, 1995; Marion et al., 1997; Hollister and Webber, 2000; Klein et al., 2005; 22 Bokhorst et al., 2011). We also observed these similar variations because of OTCs warming. In 23 addition, significant enhancements of mean monthly and mean seasonal Ta exhibited the least in 24 January and winter, but the most in April and spring, respectively. Nevertheless, mean annual Ta 25 significantly increased by 1.4 °C (P

1 alpine grassland ecosystem to experimental warming is still lacking up to now. Here our results 2 indicated that alterations of shallow Ts at 5-40 cm depths occurred at different temporal scales 3 based on experimental warming over 2009-2011 (see Table 1, Fig. 4 and 7). Maximum increases of Ts at the depths from 5 cm to 40 cm exhibited 0.4 °C ~ 2.9 °C, 1.3 °C ~ 2.3 °C, 1.2 °C ~ 1.5 °C, 4 5  $0.8 \text{ °C} \sim 1.1 \text{ °C}$  and  $0.2 \text{ °C} \sim 0.7 \text{ °C}$  in alpine swamp meadow, as well as  $0.3 \text{ °C} \sim 3.3 \text{ °C}$ ,  $1.1 \text{ °C} \sim 0.7 \text{ °C}$ 4.1 °C, 0.6 °C ~ 2.8 °C, 0.5 °C ~ 2.4 °C and 0.3 °C ~ 1.5 °C in alpine steppe at diurnal, mean daily, 6 7 mean monthly, mean seasonal and mean annual scales because of OTCs warming, respectively. 8 Further, the increase amplitude of Ts was greater in alpine steppe than in alpine swamp meadow. It 9 showed that  $T_s$  affected by OTCs treatment was notable at 5-10 cm depths, and gradually decreased with depth owing to the horizontal heat transfer. On the basis of equations (4.26-4.28) 10 11 by William and Smith (1989), the effects of OTCs on Ts at 5, 10, 20, 30 and 40 cm depths were 12 about 93.3%, 86.6%, 73.9%, 62.4% and 52.5%, respectively. Note that Ts variations might be 13 overestimated without consideration of the three-dimensional heat conduction. 14 At mean seasonal and annual scales (Table 1), SM at 10 cm depth basically decreased in two

15 alpine grasslands, but significantly declined in alpine steppe owing to OTCs warming. In addition, SS at 10 cm depth obviously increased in two alpine grasslands, and increase amplitude 16 17 significantly higher in alpine swamp meadow than in alpine steppe. Therefore, it indicated that 18 topsoil drought was significantly aggravated in alpine steppe, while salinization was more 19 significant in alpine swamp meadow than in alpine steppe under the warming scenarios. 20 Apparently, response difference of soil hydrothermal dynamics to the warming scenarios may be 21 controlled by vegetation coverage as one of the most important factors. Compared with alpine 22 swamp meadow with higher vegetation coverage (~ 87%), topsoil in alpine steppe with lower 23 vegetation coverage (~ 38%) had higher thermal conductivity and lesser heat capacity, as well as 24 was more sensitive to climate change with greater shift in soil temperature and moisture (Wang et 25 al., 2012).

#### 26 4.2 Response of FI/TI, SFTPs and ALT to experimental warming

The annual FI/TI, as a useful indicator of climate change in the cold regions, could be very useful 28 not only in evaluating distribution of permafrost and seasonally frozen ground, but in terms of 29 engineering applications (Frauenfeld et al., 2007). Up to now, FI/TI basically included air FI/TI 30 based on Ta at 1.5 m above the ground (Cheng et al., 2003; Frauenfeld et al., 2007; Jiang et al., 31 2008), and surface FI/TI determined for temperature below a surface (Zhao et al., 2008; Wu et al., 32 2013). However, FI exhibited decreasing trends while TI showed increasing trends because of 33 climate warming in recent decades. For instance, Jiang et al. (2008) detected that air FI decreased 34 by rate of 16.6-59.1 °C d decade<sup>-1</sup>, but air TI increased by rate of 19.8-45.6 °C d decade<sup>-1</sup> over 35 1966-2004 along the Qinghai-Tibet Railway. The result by Frauenfeld et al. (2007) exhibited that 36 air FI had decreased by 85.6 °C d decade<sup>-1</sup> and air TI had increased by 44.4 °C d decade<sup>-1</sup> during 37 recent decades in the Northern Hemisphere (north of 50 °N). In addition, Wu et al. (2013) revealed 38 that surface FI had decreased at a rate 111.2 °C d decade-1 and surface TI had increased at a rate of

1 125.0 °C d decade<sup>-1</sup> over the period of 1980-2007 on the central of QTP. In this paper, so-called
FI/TI referred to air FI/TI, which was calculated at 15 cm above the ground of CPs and OTCs
plots, respectively. Likewise, our results indicated that this area should be underlain by continuous
permafrost because of larger FI (1873.4 °C d) while smaller TI (886.8 °C d) under CPs treatment.
There were significant variations about FI decrease (410.7 °C d) while TI increase (460.7 °C d)
under experimental warming (Table 2), which further showed a change in permafrost status
(Frauenfeld et al., 2007).

8 As a "buffer layer" between atmosphere and permafrost, the active layer has a sensitive 9 response to climate change (Li et al., 2012a). Variation of its hydrothermal dynamics will necessarily induce the quick change in STFPs, especially of shallow soil in the permafrost regions. 10 11 The long-term monitoring results by Li et al. (2012a) revealed that the onset data of the active layer thawing was advanced by 16 days, while the onset data of freezing was delayed by 14 days, 12 13 and thawed days had increased by 30 days within the past decade along the Qinghai-Tibet 14 Highway. The work using remote sensing data found that there was a trend toward earlier onset 15 data of near-surface soil thawing by about 14 days and later onset date of freezing by approximately 10 days, as well as frozen days had decreased by 33.7 days over the period 16 1988-2007 on the QTP (Li et al., 2012b). Under the condition of Ta warming of 1.84 °C using 17 18 OTCs facility, the onset date of shallow soil thawing at 5-40 cm depths was advanced by 25 days, 19 whereas the onset data of freezing was postponed by 18 days, and as well as frozen days had 20 decreased by 43 days in wet meadow over 2008 on the central of QTP (Wang et al., 2013). Our 21 results exhibited that based on Ta warming scenarios of 1.4 °C on the central of QTP, the onset 22 dates of shallow soil thawing at 5-40 cm depths were averagely advanced by 9 days and 8 days 23 while the onset dates of freezing were delayed by 10 days and 4 days in alpine swamp meadow 24 and alpine steppe, respectively. Further, soil frozen days were significantly decreased by 28 days 25 and 16 days while thawed days were increased by 18 days and 6 days, and frozen-thawed days 26 were significantly increased by 10 days and 10 days in alpine swamp meadow and alpine steppe, 27 respectively (Table 2). It revealed the significant pattern of earlier thawing, later freezing, shorter 28 freezing period while longer thawing and freezing-thawing periods of shallow soil, especially 29 topsoil of active layer in two alpine grasslands of the permafrost regions in the central QTP owing 30 to experimental warming. Response difference of SFTPs in two types of alpine grassland mainly 31 occurred at the autumn freezing stage, which showed a trend of later freezing in alpine swamp 32 meadow than in alpine steppe. It should be resulted from a stronger preservation effort to soil heat 33 in alpine swamp meadow with higher vegetation coverage. Nevertheless, the stability of engineering infrastructure (e.g., the Qinghai-Tibet Railway or Highway) will be undoubtedly 34 35 influenced in the permafrost regions of the QTP, because of variations of geotechnical properties 36 affected by STFPs variations (Qi et al., 2006). The variation of near-surface SFTPs is also an effective and reliable indicator of permafrost 37

The variation of near-surface SFTPs is also an effective and reliable indicator of permafrost
 change on the QTP (Li et al., 2012b). Over the past several decades, permafrost has remarkably

1 degraded on the QTP due to climate warming. In any case, the increase of ALT is one of the main 2 representations of permafrost degradation. Monitoring results demonstrated that ALT had 3 increased at a rate of ~7.5 cm/yr over a period of 1995-2007 (Wu and Zhang, 2010) and 1.33 4 cm/yr in the past 30 years (Li et al., 2012a) along the Qinghai-Tibet Highway, as well as 5 ~6.3cm/yr from 2006 through 2010 along the Qinghai-Tibet Railway (Wu et al., 2012). To 6 estimate the effect of the warming scenarios on ALT in two alpine grassland ecosystems of the 7 central QTP, ALT under OTCs and CPs treatments were simulated using the empirical equation 8 (see Data analysis; r = 0.866, P<0.01). The equation was constructed using the long-term 9 monitoring data from 1998 to 2009 along the Qinghai-Tibet Highway. Our two experimental sites 10 were completely included within the study area by Li et al. (2012a). In addition, ALT were 203.0 11 cm and 212.7 cm using the empirical equation, which were very close to 200 cm and 220 cm 12 based on the pitting measurement in alpine swamp meadow and alpine steppe ecosystems under 13 CPs treatments in 2009, respectively. Therefore, the equation could be used to estimate ALT under 14 OTCs and CPs treatments at two experimental sites. However, ALT in alpine swamp meadow and 15 alpine steppe ecosystems showed the significant increases of ~6.9 cm and ~19.6 cm owing to experimental warming, respectively. In short, it was quite clear that the change in FI/TI and SFTPs 16 17 triggered by experimental warming, would surely bring about permafrost degradation in alpine 18 grassland ecosystem. Numerous studies have demonstrated that as a result of permafrost thaw, old 19 soil carbon released to atmosphere had the strong potential to serve as a large positive feedback to 20 global change (e.g., Zimov et al., 2006; Schuur et al., 2009; Koven et al., 2011; Hicks Pries et al., 21 2016). Therefore, quantification of permafrost degradation figured by ALT increase will further 22 provide the reference on estimation of carbon release under the future warming scenarios in the 23 OTP.

#### 24 5 Conclusions

In this study, response characteristics of *T*a, FI/TI, shallow soil hydrothermal dynamics, SFTPs and ALT in two types of alpine grassland ecosystem to experimental warming using OTCs facility were investigated in the permafrost regions of the Beiluhe Basin of the central QTP from 2009 to 2011. *T*a pattern exhibited a significant phenomenon of daytime warming while nighttime cooling at diurnal variation, whereas it showed an obvious increase from mean daily, monthly, seasonal and annual dynamics. Meanwhile, diurnal and annual *T*a range significantly enhanced by 5.0 °C and 0.8 °C, respectively. However, mean annual *T*a increased by 1.4 °C.

Under the condition of *T*a warming scenarios, *T*s variations were evident at 5-10 cm depths and gradually decreased with increase of profile depths, and increase amplitude of *T*s was greater in alpine steppe than in alpine swamp meadow. In general, mean annual *T*s at the depths from 5cm to 40 cm was increased by  $0.2 \sim 0.7$  °C in alpine swamp meadow and 0.3 °C  $\sim 1.5$  °C in alpine steppe. Furthermore, mean annual SM at 10cm depth decreased by 1.1 % and 0.8 %, and mean annual SS at 10cm depth significantly increased by 0.3 g L<sup>-1</sup> and 0.1 g L<sup>-1</sup> in alpine swamp meadow and alpine steppe, respectively. These showed that topsoil drought was aggravated in two

alpine grasslands, especially was significant in alpine swamp meadow. Meanwhile, topsoil
 salinization was more serious in alpine swamp meadow than in alpine steppe.

3 Further, FI was significantly decreased by 410.7 °C d while TI was significantly increased by 4 460.7 °C d. Likewise, the onset dates of shallow soil thawing at 5-40 cm depths were advanced by 5 9 days and 8 days while the onset dates of freezing were delayed by 10 days and 4 days in alpine 6 swamp meadow and alpine steppe, respectively. Moreover, soil frozen days were significantly 7 decreased by 28 days and 16 days, but thawed days were increased by 18 days and 6 days, and 8 frozen-thawed days were significantly increased by 10 days and 10 days in alpine swamp meadow 9 and alpine steppe, respectively. In addition, ALT had notable increases by about 6.9 cm and 19.6 10 cm in alpine swamp meadow and alpine steppe ecosystems, respectively. However, the response 11 differences in two types of alpine grasslands may be mainly controlled by vegetation coverage. 12 Overall, the systematically quantitative responses of freeze-thaw processes and permafrost to 13 experimental warming will greatly provide the scientific references on evaluating the 14 infrastructure stabilities (e.g., the Qinghai-Tibet Railway/Highway) and estimating the soil carbon 15 release in the permafrost regions of QTP under the future climate warming.

16 Acknowledges. We wish to thank the editor Gruber S. for the constructive comments and 17 suggestions. We thank Luo Y., Alonso-Contes, C., Pen F., Liang J., Jiang J., and Huang Y. for 18 revising manuscript. We thank Klein J. A., and Zhao X. for the help of OTCs fiberglass material. 19 This work was supported by the National Basic Research Program of China (973 program, 20 2013CBA01807), Key Project of Chinese Academy of Sciences (KJZD-EW-G03-04), the 21 Foundation for Innovative Research Groups of the National Natural Science Foundation of China 22 (41421061), the Freedom Project of the State Key Laboratory of Cryosphere Science, Cold and 23 Arid Regions Environmental and Engineering Research Institute, Chinese Academy of Sciences 24 (SKLCS-ZZ-2015-2-2), the National Natural Science Foundation of China (41171054, 40901040) 25 and the National Science & Technology Pillar Program (2014BAC05B02).

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
