# Peer review of "Response of freeze-thaw processes to experimental warming in the 1 permafrost regions of the central Qinghai-Tibet Plateau 2 3 Shengyun Chen1, 2, +, \*, Wenjie Liu3, 1, +, Qian Zhao1, 7, Lin Zhao4, Qingbai Wu5, Xingjie Lu6, 4 Shi"

_The Cryosphere, 2016_

## Referee Comment (RC1) · Anonymous Referee #1 · 23 Jun 2016

Experimental warming is widely used in ecological research to help to understand changes in carbon stocks and fractions, and the related processes resulting from climate change. However, experimental research in the open field on the response of freeze-thaw processes and the active layer of permafrost-affected soils are rare, at least to my knowledge. In this respect, the paper is a valuable contribution to the scientific discussion on the consequences of global warming. The overall quality of the paper is good and I appreciate the large number of citations in the introduction which gives a very good overview of permafrost research on the Tibet plateau. I would be happy if some more findings from ecology and soil science could be added (e.g. Baumann et al. Global Change Biology 15, 3001-3017 (2009), doi: 10.1111/j.1365-

2486.2009.01953.x, He J-S et al. New Phytologist, 170, 835–848 (2006), Geng Y et al. PLoS ONE 7 (4): e34968. (2012), doi:10.1371/journal.pone.0034968).

The results section has many sentences that consist of data rows. I suggest to shorten the text and present the data in tables.

The introduction to the discussion (p7, l37 to p8, l8) could better be integrated into the introduction section. This holds true for quite some parts of the following paragraphs of the discussion section. As mentioned in the title of the manuscript, processes of freeze-thaw cycling would play a central role. I would suggest to extend the discussion of such processes. Instead, I would leave out the objectives to estimate the release of old carbon and to evaluate the stability of Qinghai-Tibet Railway/Highway since I couldn't find any own results on this issue in the manuscript.

---

## Referee Comment (RC2) · Anonymous Referee #2 · 10 Aug 2016

This paper entitled "Response of freeze-thaw processes to experimental warming in the permafrost regions of the central Qinghai-Tibet Plateau," by Shengyun Chen et al, reports the effects of an open top chamber warming experiment over three years. The authors measured air and soil temperature, soil moisture, and soil salinity with and without chambers. The study is basically methodologically sound and the manuscript is well written and readable. The figures are also very well done. However, the scientific justification for the study is very weak, and the performed work seems more like a methodological proof of concept or a component of a larger study than a full-fledged experiment. Given how long open top chambers have been used and the purely observational nature of this study, it is not clear what this paper contributes to a larger

understanding of the effects of climate change on freeze-thaw dynamics or related hydrological and ecological processes.

A few specific suggestions for improving the manuscript:

1. This seems a little like a study in search of a response variable. Simply installing chambers and measuring if they effected soil temperature and moisture does not advance our understanding of how air temperature and soil temperature are related, nor does it provide novel insight into the ecological concerns raised by the authors in the introduction (links with vegetation, biogeochemical cycles, or permafrost degradation). Were there other parameters measured at these sites that could be leveraged to tell a more engaging story? 2. If the study is purely methodological (reworking old questions of artefacts and advantages of open top chambers), the authors should put their work in context. There are many papers that use ITEX methodology, many of which discuss in depth issues with passive warming chambers. It would be particularly relevant to discuss the effects of leaving the chambers throughout the winter when they can influence heat exchange via preferential snow accumulation. 3. The extensive use of uncommon acronyms makes the paper hard to read. What can be a shortcut for the authors becomes a stumbling block for readers. Most acronyms could and should be removed, except for a few key terms. 4. It would be helpful to develop a hypo-deductive framework around the research questions. Stating a clear hypothesis would go a long way towards justifying the study and preparing the readers to understand the implications of the results. It could also lead to a more focused analytical structure, rather than just observing differences.

---

## Author Comment (AC1) · 23 Sep 2016

**Reply comments on reviewer 1#:**

**Specific comments**

1. Experimental warming is widely used in ecological research to help to understand changes in carbon stocks and fractions, and the related processes resulting from climate change. However, experimental research in the open field on the response of freeze-thaw processes and the active layer of permafrost-affected soils are rare, at least to my knowledge. In this respect, the paper is a valuable contribution to the scientific discussion on the consequences of global warming. The overall quality of the paper is good and I appreciate the large number of citations in the introduction which gives a very good overview of permafrost research on the Tibet plateau.

Reply: Thank you very much for such a highly positive evaluation of our research. Up to now, *in-situ* experimental research using OTCs warming in the field on quantitatively systematic response of freeze-thaw processes and permafrost is extremely rare in the Qinghai-Tibet Plateau. The main objectives of this manuscript with regard to the permafrost regions of the central the Qinghai-Tibet Plateau were to quantitatively assess the response characteristics of freeze-thaw processes including air freezing/thawing index and soil freeze-thaw processes, and permafrost represented by active layer thickness in the alpine swamp meadow and the alpine steppe ecosystems to OTCs experimental warming based on the monitoring data of air temperature and soil hydrothermal condition during 2009-2011.

2. I would be happy if some more findings from ecology and soil science could be added (e.g. Baumann et al. Global Change Biology 15, 3001-3017 (2009), doi: 10.1111/j.1365-2486.2009.01953.x, He J-S et al. New Phytologist, 170, 835–848 (2006), Geng Y et al. PLoS ONE 7 (4): e34968. (2012), doi:10.1371/journal.pone.0034968).

Reply: Thanks for your suggestions. We have added these findings (He et al., 2006; Baumann et al., 2009; Geng et al., 2012) into our revised manuscript. Three references are as following:

Baumann, F., He, J., Schmidt, K., Kühn, P., and Scholten, T.: Pedogenesis, permafrost, and soil moisture as controlling factors for soil nitrogen and carbon contents across the Tibetan Plateau, Glob. Change Biol., 15, 3001–3017, doi: 10.1111/j.1365-2486.2009.01953.x, 2009.

Geng, Y., Wang, Y., Yang, K., Wang, S., Zeng, H., Baumann, F., Kuehn, P., Scholten, T., and He, J.: Soil respiration in Tibetan alpine grasslands: belowground biomass and soil moisture, but not soil temperature, best explain the large-scale patterns, PLoS ONE, 7, e34968, doi:10.1371/journal.pone.0034968, 2012.

He, J., Wang, Z., Wang, X., Schmid, B., Zuo, W., Zhou, M., Zheng, C., Wang, M., and Fang, J.: A test of the generality of leaf trait relationships on the Tibetan Plateau, New Phytol., 170, 835–848, doi: 10.1111/j.1469-8137.2006.01704.x, 2006.

3. The results section has many sentences that consist of data rows. I suggest to shorten the text and present the data in tables.

Reply: Thank you for your suggestions. We have omitted some sentences that consist of data in the Results section. However, it is still unclear that diurnal to annual variations of some variables (e.g., soil temperature) owing to OTCs experimental warming are quantitatively exhibited by Figures (e.g., Fig.4, Fig. 7). Therefore, we need to retain some sentences that include data based on Figures. Moreover, these data are difficult to be further presented clearly by Table.

4. The introduction to the discussion (p7, l37 to p8, l8) could better be integrated into the introduction section. This holds true for quite some parts of the following paragraphs of the discussion section.

Reply: Thanks for your opinions. We have integrated the part (p7, l37 to p8, l5) in the Discussion section into the part (p2, l7 to l13) in the Introduction section in the revised manuscript. However, the part (p8, l6 to l8) in the Discussion section is still retained as a guidance of OTCs experimental warming. Thus, the Introduction section is revised into:

The fifth assessment report (AR5) of the Intergovernmental Panel on Climate Change (IPCC) indicated that the global average surface temperature had increased by 0.85 (0.65 ~ 1.06) $^{\circ}$C over

the period of 1880 to 2012, and the change for the end of the 21$^{st}$ century would be likely to exceed 1.5 $^{o}$C than the average from 1850 to 1900 (IPCC, 2013). Likewise, the Qinghai-Tibet Plateau (QTP) has experienced a universal and significant warming (Wu and Zhang, 2008; Li et al., 2010; Wu et al., 2013), the recent warming rate has been greater than those for the northern hemisphere, the southern hemisphere and the world as a whole (Trenberth et al., 2007). Mean annual temperature in the QTP has increased by 0.2 $^{o}$C per decade over the past 50 years, and is expected to increase 2.6-5.2 $^{o}$C by 2100 under the A2 and B1 scenarios (Chen et al., 2013).

5. As mentioned in the title of the manuscript, processes of freeze-thaw cycling would play a central role. I would suggest to extend the discussion of such processes. Instead, I would leave out the objectives to estimate the release of old carbon and to evaluate the stability of Qinghai-Tibet Railway/Highway since I couldn't find any own results on this issue in the manuscript.

Reply: Thanks for your suggestions. In this study, we used the monitoring data of air temperature and shallow soil hydrothermal condition within and without OTCs during 2009-2011, to quantitatively assess response of freeze-thaw processes including air freezing/thawing index and soil freeze-thaw processes, and permafrost figured by active layer thickness in two typical alpine grassland ecosystems to OTCs experimental warming in the permafrost regions of the central Qinghai-Tibet Plateau. Until now, a quantitatively systematic research on the response of these variables to climate warming is extremely rare in the permafrost regions of the Qinghai-Tibet Plateau. As mentioned in the title of the manuscript, freeze-thaw processes (include freezing/thawing index and soil freeze-thaw processes) and permafrost (implied by permafrost regions) would really play the central roles. Therefore, these above variables were mainly discussed in this manuscript. In addition, we also discussed moderately the method of OTCs warming, variations of air temperature and shallow soil hydrothermal condition under the OTCs experimental warming. However, it is our hope that this work could greatly provide the scientific references on further assessing the infrastructure stability (e.g., the Qinghai-Tibet Railway/Highway), soil carbon release and alpine ecosystem health in the permafrost regions of the Qinghai-Tibet Plateau under the future climate warming. It just highlight the importance of our

research at present, and need to be intensively studied in the future. We have added some studies of soil freeze-thaw processes in the Discussion section as following:

The near-surface soil freeze-thaw processes is an important indicator of climate change, which is primarily controlled by air temperature (Wang et al., 2015). Using data from 636 meteorological stations across China, Wang et al. (2015) found that the onset date of near-surface soil freezing was delayed by about 5 days, and the frozen days decreased by about 10 days over the period 1956-2006.

In addition, we have also merged the part (p10, l33 to l36) and the part (p11, l18 to l23) in the Discussion section, and revised into:

However, numerous studies have showed that changes in not only freeze-thaw processes but also permafrost triggered by climate warming, strong affected vegetation productivity, soil physicochemical properties, carbon feedback and engineering infrastructure, etc. (e.g., Grogan et al., 2004; Qi et al., 2006; Zimov et al., 2006; Kreyling et al., 2008; Wang et al., 2009; Chen et al., 2012a; Hicks Pries et al., 2016). Therefore, it is very necessary for us to further conduct the systematic monitoring research that extremely limited in the permafrost regions of the QTP in the future.

---

## Author Comment (AC2) · 23 Sep 2016

**Reply comments on reviewer 2#:**

**Specific comments**

This paper entitled "Response of freeze-thaw processes to experimental warming in the permafrost regions of the central Qinghai-Tibet Plateau," by Shengyun Chen et al, reports the effects of an open top chamber warming experiment over three years. The authors measured air and soil temperature, soil moisture, and soil salinity with and without chambers. The study is basically methodologically sound and the manuscript is well written and readable. The figures are also very well done. However, the scientific justification for the study is very weak, and the performed work seems more like a methodological proof of concept or a component of a larger study than a full-fledged experiment. Given how long open top chambers have been used and the purely observational nature of this study, it is not clear what this paper contributes to a larger understanding of the effects of climate change on freeze-thaw dynamics or related hydrological and ecological processes.

Reply: Thank you very much for the constructive comments and suggestions which help to improve this manuscript. Up to now, a quantitatively systematic assessment is extremely lacking about the response of freeze-thaw processes including freezing/thawing index and soil freeze-thaw processes, and permafrost figured by ALT to climate warming in the permafrost regions of the Qinghai-Tibet Plateau. In this study, we used the monitoring data of air temperature and shallow soil hydrothermal condition within and without OTCs during 2009-2011, to quantitatively assess the response characteristics of freeze-thaw processes and ALT in the alpine swamp meadow and the alpine steppe ecosystems to OTCs experimental warming in the permafrost regions of the Beiluhe Basin of the central Qinghai-Tibet Plateau. Therefore, our study is to mainly reveal the quantitative effect of OTCs experimental warming on freeze-thaw processes and permafrost represented by ALT in the Qinghai-Tibet Plateau, but is not a methodological proof of concept or a component of a larger study. In order to clearly reflect the above main objectives of this study, we have revised the Abstract and Introduction sections. In addition, we also added some explanation

in the Introduction section as following:

1) Based on the long-term monitoring data in the permafrost regions of the QTP, some studies have revealed respectively the indubitable facts about changes in FI/TI, soil freeze-thaw processes and ALT owing to climate warming (e.g., Jiang et al., 2008; Li et al., 2012a; Wu et al., 2015).

2) Under a certain increase of air temperature using OTCs facility, we attempted to mainly answer the following questions: 1) how experimental warming affected freeze-thaw processes, and 2) to what extent the warming affected permafrost represented by ALT in the alpine swamp meadow and the alpine steppe ecosystems.

A few specific suggestions for improving the manuscript:

1. This seems a little like a study in search of a response variable. Simply installing chambers and measuring if they effected soil temperature and moisture does not advance our understanding of how air temperature and soil temperature are related, nor does it provide novel insight into the ecological concerns raised by the authors in the introduction (links with vegetation, biogeochemical cycles, or permafrost degradation). Were there other parameters measured at these sites that could be leveraged to tell a more engaging story?

Reply: Thanks for your comments. In this study, using the monitoring data of air temperature and shallow soil hydrothermal condition with and without OTCs during 2009-2011, a systematic response of freeze-thaw processes including freezing/thawing index and soil freeze-thaw processes, and permafrost represented by ALT in the alpine swamp meadow and the alpine steppe ecosystems to OTCs experimental warming were quantitatively assessed in the Qinghai-Tibet Plateau. Therefore, freezing/thawing index, soil freeze-thaw processes and permafrost were the main variables in this manuscript. In order to describe changes in these variables and their effect on alpine ecosystem, carbon feedback and engineering infrastructure and so on under climate warming, we have cited the large number of references in the Introduction section. From our monitoring data, we could clearly understand changes in soil temperature and soil moisture content owing to OTCs experimental warming (e.g., Table 1, Fig. 4-7). Because air temperature

and soil temperature were not the main response variables in this study, we did not present their relationship in the Introduction section. However, we moderately discussed them and their relationship as microenvironment element in the Discussion section.

Actually, our original research object using OTCs facility is to reveal the response of plant phenology and freeze-thaw processes to experimental warming. Thus, we also measured plant phenology in the alpine swamp meadow and the alpine steppe. However, it is very regretful that this parameter could not be leveraged to tell a more engaging story in this manuscript. It is our hope that other parameters (e.g., vegetation productivity, soil carbon storage and greenhouse gas emission, etc.) within and without OTCs in future.

2. If the study is purely methodological (reworking old questions of artefacts and advantages of open top chambers), the authors should put their work in context. There are many papers that use ITEX methodology, many of which discuss in depth issues with passive warming chambers. It would be particularly relevant to discuss the effects of leaving the chambers throughout the winter when they can influence heat exchange via preferential snow accumulation.

Reply: Thanks for your suggestions. This study is not purely methodological statement of OTCs facility. We used the monitoring data of air temperature and shallow soil hydrothermal condition within and without OTCs, to quantitatively assess the response characteristics of freeze-thaw processes and permafrost to OTCs experimental warming in the alpine swamp meadow and the alpine steppe ecosystems in the central Qinghai-Tibet Plateau. Therefore, OTCs facility designed in accordance with standards of the ITEX was simply discussed advantages and disadvantages in the Discussion section. However, this passive warming facility has been widely used in the remote and no-electric regions including the Arctic, Antarctica and Qinghai-Tibet Plateau, which is also discussed about advantages and disadvantages (e.g., Wan et al., 2002; Bokhorst et al., 2013). In the revised manuscript, we have added descriptions of snow cover that related to soil thermal condition and freeze-thaw processes, and that is affected by OTCs facility in the Introduction and Discussion sections as following:

1) Introduction section:

Soil freeze-thaw processes can be significantly affected by complexly environmental factors including climate, snow cover, microtopography, vegetation and hydrology, etc. (Zhang, 2005; Minke et al., 2009; Guglielmin et al., 2012; Wang et al., 2012; Wang et al., 2015).

2) Discussion section:

However, snow cover has a significant influence on the soil thermal condition due to the insulation effect, which can result in an increase of mean annual soil temperature in the continuous permafrost regions (Zhang, 2005). It may be an important contributor to the near-surface soil freeze-thaw processes (Wang et al., 2015). In addition, deeper snow can be trapped in OTCs facility (Bokhorst et al., 2013). Therefore, the effect of snow cover on soil freeze-thaw processes should be taken into full account under OTCs warming in future.

Meanwhile, we added one reference:

Zhang, T.: Influence of the seasonal snow cover on the ground thermal regime: An overview, Rev. Geophys., 43, RG4002, doi:10.1029/2004RG000157, 2005.

However, it is very regretful that we do not measure snow cover in present. We will monitor the variable in future.

3.  The extensive use of uncommon acronyms makes the paper hard to read. What can be a shortcut for the authors becomes a stumbling block for readers. Most acronyms could and should be removed, except for a few key terms.

Reply: Thank you for your suggestions. We have removed some uncommon acronyms except for some key and common terms such as FI/TI (freeze/thaw index), OTCs (open-top chambers), CPs (control plots), ALT (active layer thickness) and QTP (Qinghai-Tibet Plateau).

4.  It would be helpful to develop a hypo-deductive framework around the research questions. Stating a clear hypothesis would go a long way towards justifying the study and preparing the readers to understand the implications of the results. It could also lead to a more focused analytical structure, rather than just observing differences.

Reply: Thanks for your suggestions. We have added some sentences in the Discussion section:

Under a certain increase of air temperature using OTCs facility, we attempted to mainly answer the following questions: 1) how experimental warming influenced freeze-thaw processes, and 2) to what extent the warming affected permafrost represented by ALT in two typical alpine grassland ecosystems.